# Influence of Tree Vegetation and The Associated Environmental Factors on Soil Organic Carbon; Evidence from "Kulon Progo Community Forestry," Yogyakarta, Indonesia

**Siswo** [1,2], **Hojin Kim** [1], **Jeongeun Lee** [1] **and Chung-Weon Yun** [1,*]

[1] Department of Forest Science, Kongju National University, Yesan-gun 32439, Chungnam, Republic of Korea
[2] Office for Standard Implementation of Environment and Forestry Instruments, Solo City 10270, Indonesia
* Correspondence: cwyun@kongju.ac.kr; Tel.: +82-41-330-1305 or +82-104-312-5745

**Abstract:** This study aimed to assess the influence of tree vegetation and some environmental factor on soil organic carbon (SOC), which is part of soil organic matter (SOM). Vegetation survey and soil sampling were carried out in five stand types in the protected forest of Kulon Progo Community Forestry, including *Pinus* (PN), *Aleurites-Swietenia* (AS), *Swietenia-Acacia* (SA), *Melaleuca-Acacia* (MA) and *Tectona-Dalbergia* (TD). Tree vegetation composition and characteristics (diversity, basal area, density, canopy height and canopy cover), SOC and SOM were analyzed using some comparative analyses. The influence of tree vegetation characteristics and environmental factors related to topographic, edaphic and anthropogenic aspects on SOC was performed by employing principal component analysis (PCA) and redundancy analysis (RDA). Our result confirmed that species composition among stand types was significantly different. Characteristically, PN was relatively close to MA, having similarities in canopy cover, canopy height and basal area. While AS, SA and TD were relatively similar to each other in diversity, basal area, density and canopy cover. Moreover, PN and MA similarly hold less SOC and SOM compared to TD, while AS and SA showed not significantly different from TD. RDA confirmed the high influence of tree vegetation, where the most influencing factor for SOC and SOM was an interaction among canopy cover, canopy height and below-stand utilization, where canopy cover was directly proportional to canopy height and increased with decreasing below-stand utilization. We concluded that in relatively small variations of environmental factors, selecting dense-canopy trees and adaptive management of below-stand utilization promised SOC sequestration and storage. Our findings provide fundamental information for maximizing the potential of forest carbon to meet the global payments for ecosystem services and contribute to low-carbon development strategies and emission reduction.

**Keywords:** tree vegetation; soil organic carbon; soil organic matter; protected forest; community forestry

## 1. Introduction

Soil Organic Carbon (SOC) is one of the most important factors determining soil quality, nutrient availability, plant growth and productivity [1] and corresponds to C sequestration potential [2,3]. The availability of SOC provides a significant effect on the global carbon cycle [4] and holds an important role in the response of the biosphere to increased atmospheric carbon dioxide [5,6]. Therefore, the perception of the sequestration potential of SOC is important knowledge to increase public attention and participation in achieving greenhouse gas (GHG) emission reduction targets of many countries.

Carbon stored in soil is about two to three times greater compared to that in living vegetation and is becoming the most dominant component of the terrestrial carbon pool [7–9]. However, the sequestration of SOC varies as the variations occur in biotic characteristics, environmental factors and anthropogenic activities [10–13]. Accordingly, variations of vegetation types in certain local areas may lead to the difference in SOC [14–16], although

the influences are also related to differences in soil conditions such as land use [17], climate, soil physicochemical [18] and topographical features [11,12]. Meantime, the individual effects of several environmental factors, such as topography and slope, are possible in controlling SOC by displacing soil particles and soil organic matter (SOM) [19].

Variation of vegetation commonly affects SOC [15] related to input of soil organic matter, decomposition process and alterations [11,20]. Therefore, forests provide important roles in the carbon balance reflected by the amount of carbon stored in the vegetation biomass, SOC and the amount circulated per year [21]. In this process, forests with different vegetation (tree species) also influence the microenvironment and soil characteristics [22] and provide different amounts and quality of litter, root exudates and soil properties, which possibly influence the soil microbial community [23]. Some research showed that land use and vegetation-change affect biomass and vegetation carbon, temperature, humidity, erosion and soil fertility and lead to lower levels of SOC, nitrogen and many essential nutrients [13,21,24,25]. Research conducted by Agus et al. [26] showed that the re-vegetation program improved SOC, nitrogen and soil pH in West Kalimantan. Some studies in China also revealed similar results that re-vegetation programs improve SOC [3,24,27,28]. Meanwhile, different tree species or stand types in Merbabu Mt. National Park Central Java also significantly affect SOC concentration [29].

In the protected forest of Kulon Progo Community Forestry and many other areas of Community Forestry in Indonesia, studies on environmental/ecosystem services were commonly limited to ecotourism potential [30] and some information on living-trees carbon assessment [31]. Carbon storage potential involved SOC as the ecosystem service was barely discussed. This is pitiable given the large potential of carbon stored in soil [7–9]. In addition, in a revegetated forest such as the study site, SOC is also one of the essential indicators to assess the success of forest revegetation [3,24,27,28]. Moreover, with increasing interest in global payments for ecosystem services such as carbon trading mechanisms [32], studies in SOC will provide input for a more comprehensive carbon service assessment of a forest area. As regulated in Presidential Regulation no 98 of 2021, followed by Ministerial Decree of Environment and Forestry no 21 of 2022, forest managers potentially receive payments for carbon services [33,34] in addition to receipts from non-timber forest products [35,36] and tourism services [30,37,38]. In this regard, carbon stored in soil could be considered in the forest carbon stock calculation, not just a soil property. Therefore, a study on SOC related to tree vegetation and environmental conditions needs to be carried out in order to provide more information and consideration in the evaluation of the forest carbon potential.

This study investigated tree vegetation in various stand types and the influence on SOC by considering environmental factors related to topographic, edaphic and anthropogenic aspects. Tree vegetation composition and characteristics, as well as the SOC content among five stand types, i.e., *Pinus* (PN), *Aleurites-Swietenia* (AS), *Swietenia-Acacia* (SA), *Melaleuca-Acacia* (MA) and *Tectona-Dalbergia* (TD), were compared. The main influencing factors on SOC were also assessed. We hypothesized that different tree vegetation compositions and characteristics, as well as the associated environmental factors, would contain different SOC concentrations and stock. We used the Protected Forest of Kulon Progo Community Forestry as a study area since it provided an excellent context for the study, which fulfilled the analytical framework of our study. The study site was a landscape with randomly distributed mixed tree vegetation, which can be classified into several stand types to compare. In addition, Kulon Progo Community Forestry is a famous successful Community Forestry in Indonesia. Therefore, the obtained information will provide valuable lessons and exemplifications for other community forestry programs in order to enhance the potential for community welfare by maximizing forest carbon potential to meet the global payments for ecosystem services and to contribute to low carbon development strategies and emission reductions as targeted in the Nationally Determined Contributions (NDC).

## 2. Materials and Methods

### 2.1. Study Site

A field investigation was conducted in the protected forest of Kulon Progo Community Forestry located in the Kokap Sub-district, Kulon Progo Region, Special Territory of Yogyakarta, Indonesia (Figure 1). This is a part of the Kulon Progo state forest, where about 196.8 ha of the forest was managed by the local community by adopting a social forestry program through the Community Forestry scheme. The area of the Community Forestry scheme with protected status reached about 114 ha, almost 60% of the total area of Community Forestry in the region [31,39]. Topographically, Kulon Progo Community Forestry is located on one of the hillsides on the line of Menoreh Hills [31], with variations in slope between 5% and 42% and elevations ranging from 100 to 450 masl [39,40]. The soil type was latosol [41], as a common feature of all the forest areas in the Kulon Progo region [39]. In the climate aspect, the area of Kulon Progo Community Forestry is categorized as C type of climate based on Schmidt Ferguson classification with high rainfall (>2500 mm/year. 4.5–6 dry months and 6–7.5 wet months) and 26.9–30 °C of average temperature [39,42,43].

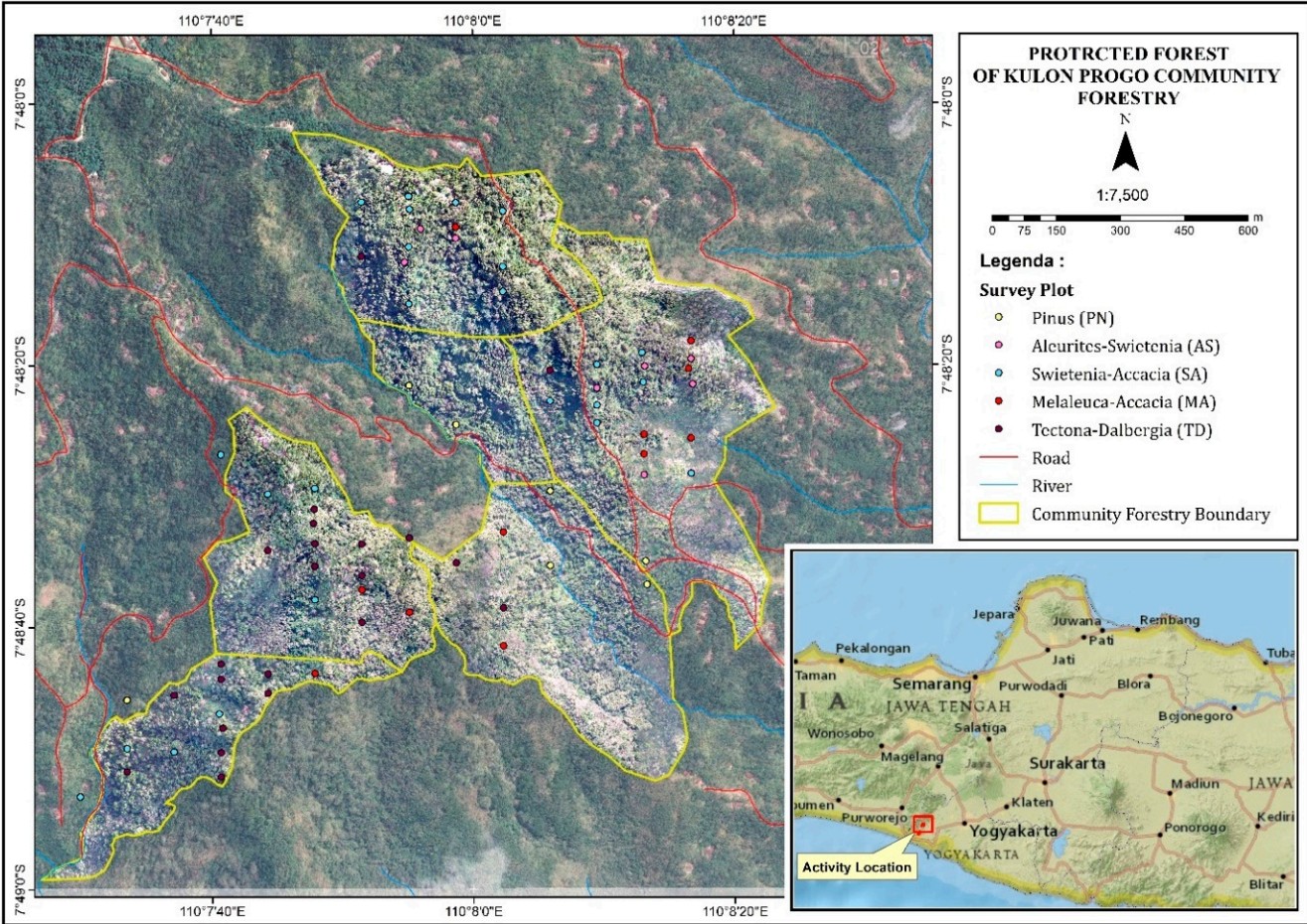

**Figure 1.** Map of the study site. Stand type and abbreviation; PN = Pinus stand, AS = Aleurites—Swietenia stand, SA = Swietenia-Acacia stand, MA = Meulaleuca-Swietenia stand, TD = Tectona-Dalbergia stand.

The protected forest of Kulon Progo Community Forestry has suffered massive deforestation since 1970 and peaked in 1998 [44]. However, as rehabilitation efforts continue, this area has now been successfully reforested. The Kulon Progo Community Forestry can be seen as a successful participatory forest rehabilitation story that has been officially carried out since 2003 [45]. The successful of the rehabilitation program was also continuously

strengthened by the designation of this area as a protected forest in 2007 [46]. With this regard, the utilization of environmental services then became very important to support community welfare in order to maintain and increase community awareness of protecting and preserving the forests. Some parts of this area have provided their environmental services (especially tourism) as an alternative to benefit sharing for the community. However, most other parts still need to be explored and utilized for potential environmental services or other ecosystem services such as carbon sequestration, water retention and the provision for non-timber forest products.

Currently, the forest cover of the Protected Forest of Kulon Progo Community Forestry is a mixed-forest dominated by trees such as *Tectona grandis*, *Dalbergia latifolia*, *c*, *Acacia mangium*, *Pinus merkusii*, *Melaleuca Leucadendra, Aleurites molucana* and *Eucalyptus* sp. [40,45]. However, the distribution of tree species was not patterned in a block management scheme, as the afforestation was carried out according to community preferences and seedling availability. Consequently, the level of stand density among utilization permits called "*andil*" also varies. This is also related to the community's dependence on below-stand utilization at the initial forest development, especially in the spots where the utilization of environmental services (tourism) has not been established.

### 2.2. Vegetation Sampling and Survey

We selected sample sites based on stand classification obtained from aerial photography confirmed by quick field-check through 72 plots as reported by Siswo et al. [40]. In this regard, tree vegetation in the protected forest of Kulon Progo Community Forestry has been classified into 5 stand types with various levels of canopy coverage (Table 1). In this study, we conducted a more detail vegetation survey through a quantitative method within the existing 72 survey plots, i.e., PN = 7 plots, AS = 8 plots, SA = 24 plots, MA = 11 plots and TD = 22 plots (Table 2). Plots were nested quadratic [47]. The nested quadratic plot consisted of a 20 × 20 m plot for the tree layer (mature trees; ≥20 cm diameter at breast height), a 10 × 10 m plot for the sub-tree layer (pole: young trees; >10 cm diameter at breast height) and a 5 × 5 m plot for saplings (small trees, 2–10 cm diameter at breast height).

**Table 1.** Stand types in the protected forest of Kulon Progo Community Forestry.

| No | Stand Type | Species Dominant | Canopy Cover (%) |
|---|---|---|---|
| 1 | PN | *Pinus Merkusii* | 68.43 |
| 2 | AS | *Aleurites miolucana*, *Swietenia macrophylla* | 75.50 |
| 3 | SA | *Swietenia macrophylla*, *Acacia auriculiformis*, *Tectona grandis*, *Dalbergia latifolia* | 74.29 |
| 4 | MA | *Melaleuca Leucadendra*, *Acacia auriculiformis* | 59.71 |
| 5 | TD | *Dalbergia latifolia*, *Tectona grandis* | 85.43 |

Source: Siswo et al. [40]. Note: PN = *Pinus* stand, AS = *Aleurites-Swietenia* stand, SA = *Swietenia-Acacia* stand, MA = *Meulaleuca-Acacia* stand, TD = *Tectona-Dalbergia* stand.

In the current study, we recorded tree vegetation characteristics by taking note of species names, counting the number of individuals and measuring the diameter at breast height (dbh). The number of individuals and diameter are the common fundamental measure in a quantitative survey related to the density and basal area reflecting species domination [48,49]. We also estimated the maximum tree height (canopy height) and canopy coverage. In addition to vegetation data, we simultaneously took some environmental factors. The influence of vegetation on soil is often related to many associated factors such as climatic, topographic, edaphic and anthropogenic factors [11,17]. We did not include climatic data as the study site was known to lie on a single stretch/area with the same temperature and annual rainfall [39]. Topographic factors, including location, altitude, slope gradient and aspect, were cited from Siswo et al. [40], which were generated from a global positioning system (GPS) and map. Slope position and bare rock were also noted based on observation. Furthermore, the estimation of below-stand utilization as an anthropogenic factor was also noted. We also counted other factors related to below-stand

utilization, including distance from road (DR) and distance from streams (DS) generated from Google Earth. Furthermore, some edaphic factors were also collected through soil samples. These factors were commonly considered as associated with vegetation [50] and some edaphic conditions [12,13,51].

**Table 2.** Description of experimental design.

| No | Survey Variables | | Description | Method/Tools |
|---|---|---|---|---|
| 1 | Existing information on vegetation classification [40] | | | |
| | - | Mapping | Aerial photography | Drone |
| | - | Plot sampling | Seventy-two quadratic plots (20 × 20 m single quadratic plot), determined and placed randomly on the map | Stratified random sampling |
| | - | Tree species investigation | Tree species identification and coverage estimation as a field check for the forest cover visually investigated on the map (72 plots). | Map-based visual investigation and field qualitative survey |
| | - | Estimation of canopy cover | Canopy density was measured as a field check for the percentage of forest cover. | Spherical densiometer |
| | - | Environmental factors | General environmental factors (altitude, slope position and aspect) | Global positioning system (GPS) and topographic map |
| 2 | Tree vegetation, more detail environmental factors and SOC investigation (current Study) | | | |
| | - | Plot sampling: | | |
| | (a) | For tree vegetation composition | 72 existing plots determined by Siswo et al. [40]: PN = 7 plots, AS = 8 plots, SA = 24 plots, MA = 11 plots and TD = 22 plots | Purposive sampling |
| | (b) | For tree vegetation characteristics and soil sample | Seven selected plots from each stand type | Purposive sampling |
| | - | Tree species investigation | Species identification, density, diameter, height (tree layer, sub-tree layer and shrub layer) | Quantitative survey |
| | - | SOC investigation | Soil analysis (SOC, SOM, bulk density and soil depth) | Soil sampling |
| | - | Environmental factors: | | |
| | a. | Topographic | Altitude, slope and aspect | Siswo et al. [40] |
| | b. | Edaphics | Bare rock, soil pH, bulk density and soil texture | Observation and soil sampling |
| | c. | Anthropogenic | Below-stand utilization, distance from road and distance from river | Observation and desk study |

*2.3. Soil Samples Collection and Laboratory Analysis*

Soil samples were collected and analyzed to investigate SOC, SOM and other related soil properties, including bulk density (BD), soil acidity (pH) and texture (silt, clay, sand). SOC is an integral part of SOM, where the SOC value is often simply calculated as 58% of SOM [11,28,52]. Meanwhile, BD is a factor determining the amount of SOC in the soil [4,13,53]. Soil pH and soil texture (percentage of silt, sand and clay) are also recognized widely as factors associated with SOC and other soil fertility indicators [13,54–57]. For

this analysis, soil samples were taken purposively to represent each stand type with seven replications. In total, we collected 35 soil samples from 5 stand types. Soil samples were collected from the topsoil layer (10–15 cm) after removing herbs and the overlying litter. We collected soil samples only from the topsoil layer because most of the variation in SOC occurs in this layer [5,13] especially related to vegetation effects [29]. Soil samples were analyzed at the Soil Laboratory of the Faculty of Agriculture, Sebelas Maret University. Soil analysis procedure referred to PPT [58] where SOC and SOM content (concentration) were analyzed using the Walkley & Black method, Soil texture was analyzed using the pipette method, BD was examined through the volumetric method, while pH was analyzed using a pH meter at a 2.5:1 (soil:water).

*2.4. Data Processing*

We characterized tree vegetation by species abundance measures, including basal area, density and important value (IV), to confirm species domination [48,49,59]. We computed those parameters at the plot level and also averaged them to provide values per stand type. Furthermore, we also considered the species diversity reflected by the Shannon diversity index [48,59]. In addition, the estimation of the percentage of canopy cover (canopy density), as reported by Siswo et al. [40], was also included to characterize the stand types. Canopy density is closely related to canopy gap and shade influencing micro climate [49,60] and soil condition [22,51].

The abundance measures and diversity were calculated by using the following formulas [48,49,59]:

$$\text{Stand density} \left(\text{individual ha}^{-1}\right) = \frac{n}{A} \tag{1}$$

$$\text{Individual Basal area} \left(BAi\right) = \frac{\pi(D \times 0.1)^2}{4} \tag{2}$$

$$\text{Basal area per ha} \left(BA\right) = \frac{\sum BAi}{A \, (ha)} \tag{3}$$

$$\text{Relative density} = \frac{nj}{\sum nj} \times 100 \tag{4}$$

$$\text{Relative basal area} = \frac{BAj}{\sum BAj} \times 100 \tag{5}$$

$$\text{IV} = \frac{Relative \, density + Relative \, basal \, area}{2} \tag{6}$$

$$H' = -\sum \left(\frac{n.i}{N}\right) \ln \left(\frac{n.i}{N}\right) \tag{7}$$

where $n$ is the number of species, $A$ is the total area of plots (ha), BAi is the basal area of $i$th species of j species, $nj$ is the number of j species, $BA$ is basal area, $BAj$ is the $BA$ of $j$ species, $H'$ is the Shannon diversity index, N is the total number of individuals and $n.i$ is the number of $i$th species.

For the analysis of soil organic carbon, we included the concentration and stock of SOC. In addition, we also considered the concentration and stock of SOM in the analysis and discussion on SOC because SOC is an integral part of SOM. We calculated SOC stock using the following equation [24,53]:

$$\text{SOCs} \left(\text{Mg ha}^{-1}\right) = \text{SOCc} \% \times \text{BD} \times \text{SD} \times 100 \tag{8}$$

where SOCs is soil organic carbon stock (Mg ha$^{-1}$ or ton/ha), SOCc is soil organic carbon concentration (%), BD is bulk density (g/cm$^2$), SD is soil depth (cm), 100 is the conversion factor from g/cm$^2$ to ton/ha.

Equation (8) was also applied to calculate SOM stock from SOM concentration data.

We arranged species abundance per plot for a comparison analysis of tree vegetation characteristics among stand types, including diversity index, density, basal area, maximum height (canopy height) and canopy coverage. Furthermore, we set SOC data, including the concentration and stock of SOC and SOM, according to stand types to assess the variation of SOC and SOM in different stand types. We also arranged the data for ordination analysis, including response/dependent variables (the concentration and stock of SOC and SOM) and the explanatory variables, including vegetation factors (Shannon diversity index, density, basal area, max height (canopy height) and canopy cover), topographic factors (elevation, slope position, slop gradient and bare rock), edaphic factors (pH, BD, clay, sand and silt) and factors related to anthropogenic (below-stand utilization, distance from road and distance from streams).

*2.5. Statistical Analysis*

We confirmed the difference among stand types by comparing tree species composition (species and the IV) using multi-response permutation procedures analysis (MRPP) in PC-ORD 7. This analysis is a nonparametric analysis ignoring distributional assumption, which is appropriate for ecological community data [59]. Furthermore, we compared tree stand characteristics, including diversity index, density, basal area, maximum height (canopy height) and canopy coverage using a 1-way analysis of variance (ANOVA) followed by a posthoc test using the Bonferroni method. ANOVA has been applied in many studies to compare species and environmental characteristics [13,25,29,50]. Kolmogorov–Smirnov and Shapiro–Wilk tests were applied to check normality data. We also performed data log-transformation to improve normality. In case the data does not meet the requirements for parametric analysis, we employed non-parametric analysis, i.e., Kruskal–Wallis test followed by Mann–Whitney U-tests for pairwise comparison [61]. Furthermore, we also employed ANOVA and the related analysis to compare SOC among stand types. ANOVA and Kruskal–Wallis tests, as well as the posthoc test, were performed using Statistical Product and Service Solution (SPSS) software version 25.0.

Considering many associated factors with SOC and SOM [12,13,62], we further analyzed the influence of tree stand characteristics and other associated factors on SOC. We include vegetation factors (Shannon diversity index, density, basal area, max height (canopy height) and canopy cover), topographic factors (elevation, slope position, slope gradient and bare rock), edaphic factors (pH, BD, clay, sand and silt) and factors related to anthropogenic (below-stand utilization, distance from road and distance from streams) as the explanatory variables and the concentration of SOC and SOM as the response/dependent variables. For this analysis, because of the unbalanced nature of many variable combinations in the data sets, we employed an exploratory pattern association type approach (ordination method) by using a redundancy analysis (RDA) [12,13,62] run in PC-ORD 7 [63]. Moreover, variance partitioning was also performed to show the proportion or partial effect of the explanatory variables.

RDA is a multivariate (multi-response) multiple regression [64], so it is possible to include more than 1 response/dependent variable [12,13,62]. However, like regression analysis, independent variables in RDA should have only a handful of variables and not strongly correlate with each other [63]. Therefore, prior to running the RDA, we summarized/grouped the explanatory variables by applying principle component analysis (PCA) to handle the *multicollinearity* [65]. We further used the score values of the grouped/summarized variables as the explanatory variable in RDA. By running PCA, we also intended to find out the interrelation among the independent variables in influencing SOC and to assess the nature of their relationship. PCA was run in SPSS software as this software provided a detail explanation of the summarized/grouped variables [66].

## 3. Results

*3.1. Variations in Tree Vegetation Composition and Characteristics*

3.1.1. Species Diversity

Our study found 28 tree species with diameters >2 cm belong to 13 families from a total of 72 vegetation survey plots. A total of 16 species were found in the tree layer/mature tree, 17 species in the pole 22 species in the shrub layer (Appendix A Table A1). Each stand type was dominated by specific species resulting in significant differences in species composition (Appendix A Table A2). As shown in Appendix A Table A1, Pinus merkusii at the tree layer in PN was greatly dominant with more than 70% of IV, and the total IV of the other seven species reached only 29.87%. Meantime, Aleurites molucana in AS, Acacia mangium in SA, Melaleuca leucadendra in MA and Dalbergia latifolia in TD showed only 52.39%, 30.78%, 39.80% and 36.89% of IV, respectively. In these stands, distribution among the two most dominant species was more even though these species similarly hold more than 65% of IV in the composition. Consequently, the value of H' in PN differed from other stands. At the plot level, PN showed significantly lower values of H' than those of other stands (Table 2). In the smaller growth levels/layers, i.e., the sub-tree layer (pole) and shrub layer (sapling), tree species distribution was greater, even resulting in similar values of H'.

3.1.2. Structural Characteristics

We found significant differences in structural characteristics among stand types (Table 3). At the tree layer, PN significantly showed higher density than other stand types. Nevertheless, they were not significantly different in terms of basal area. Inversely, in the sub-tree layer, AS, SA, MA and TD stands significantly exhibited higher densities compared to PN. Uniquely, they were equal in basal area (Table 3). Furthermore, we found different patterns in the shrub layer where the entire stand types had equal tree density. Basal area was also mostly similar, with only PN having a smaller basal area than other stand types. Moreover, in addition to density and basal area, we found significant differences in canopy cover and height. The smallest canopy cover was in MA and PN, which is significantly distinct from the TD stand. Meanwhile, max height (canopy height) was mostly similar, except between the PN and AS stands, where the canopy height in AS was significantly lower than PN.

**Table 3.** Plot-level tree species diversity and structural characteristics of tree species in the protected forest of Kulon Progo Community Forestry.

| Tree Characteristics | Mean/Mean Rank | | | | | F/p Value | Test Used |
| --- | --- | --- | --- | --- | --- | --- | --- |
| | PN | AS | SA | MA | TD | | |
| N | 7 | 7 | 7 | 7 | 7 | | |
| Diversity index (T) | 0.48/10.14 a | 0.54/13.86 b | 1.11/24.43 b | 1.07/27.14 b | 0.75/14.43 b | 0.006 | KW |
| Diversity index (ST) | 0.39/12.07 a | 0.72/19.71 a | 0.84/23.36 a | 0.79/17.93 a | 0.67/16.93 a | 0.326 | KW |
| Diversity index (S) | 0.59/20.29 a | 0.40/16.57 a | 0.82/24.14 a | 0.34/13.86 a | 0.37/15.14 a | 0.284 | KW |
| Basal Area (T) | 27.61 a | 17.56 a | 15.94 a | 19.91 a | 23.55 a | 0.422 | AN |
| Basal Area (ST) | 3.98/11.64 a | 6.45/18.43 a | 7.26/21.57 a | 5.06/17.50 a | 5.68/20.86 a | 0.39 | KW |
| Basal Area (S) | 0.73/7.71 a | 2.76/18.57 b | 4.89/26.29 b | 2.83/18.50 ab | 2.97/18.93 b | 0.019 | KW |
| Density (T) | 375 a | 204 b | 232 b | 154 b | 211 b | 0.001 | AN |
| Density (ST) | 2007.79 a | 343/16.71 b | 528/22.57 b | 400/21.07 b | 429/21.86 ab | 0.004 | KW |
| Density (S) | 886/15.07 a | 1028/16.21 a | 1842/27.14 a | 1142/16.07 a | 971/15.50 a | 0.179 | KW |
| Max height | 28.14 a | 19.29 b | 21.29 ab | 21 ab | 27.14 a | 0.005 | AN |
| Canopy coverage | 68.43/12.79 a | 75.5/19.21 abc | 74.29/19.21 abc | 59.71/11.43 ab | 85.43/27.36 c | 0.029 | KW |

Note: Stand type; PN = *Pinus* stand, AS = *Aleurites-Swietenia* stand, SA = *Swietenia-Acacia* stand, MA = *Meulaleuca-Acacia* stand, TD = *Tectona-Dalbergia* stand. Growth level; T = tree layer, ST = sub-tree layer, S = shrub layer. Test used: KW = Kruskal wallis test, AN = Annova. N = number of samples. Different letters (a,b,c) demonstrate significant differences between plot groups. Computed using alpha 0.05.

*3.2. SOC Variation and the Influencing Factors*

3.2.1. SOC Variation According to Vegetation Types

Values of SOC content varied among sample plots where both the concentration and stock of SOC were directly proportional to those of SOM (Figures 2 and 3). This is inherent to the common knowledge that SOC is part of SOM for about 58% [11,25,52]. In addition, Figures 2 and 3 also showed that the SOC stock in this study was directly proportional to the concentration due to weak variation of BD among sample plots. Furthermore, our study found significant differences in SOC among stand types for both concentration and stock (Figure 2a,c). The PN and MA stands significantly exhibited lower SOC than TD. However, they were not significantly different compared to AS stand and SA. On average, the SOC concentrations in PN and MA were only 1.05% and 0.98%, with stocks of 18.08 ton/ha and 16.94 ton/ha, respectively. Meanwhile, TD showed the highest SOC among the entire stand types (1.72% SOC concentration and 30.82 ton/ha SOC stock.

3.2.2. SOC Variation between Plots

On average, soil in the protected forest of Kulon Progo Community Forestry contained approximately 1.25% of SOC with a storage potential of 21.93 tons/ha, which is about 58% of SOM (2.18% and 38.11 tons/ha for the concentration and stock, respectively). From total soil sample plots, the SOC and SOM concentrations varied from 0.67%–2.35% and 1.15%–4.05%, respectively (Figure 2a). Meanwhile, the SOC and SOM stock successively ranged from 11.26 to 45.12 ton/ha and 19.32–77.79 ton/ha (Figure 2b). The coefficient variations of SOC concentration, SOM concentration, SOC stock and SOM stock were consecutively 34.95%, 35.16%, 36.35% and 36.80%. Moreover, we found that the maximum values of SOC and SOM were in some plots of TD stand type (Figure 2a,b).

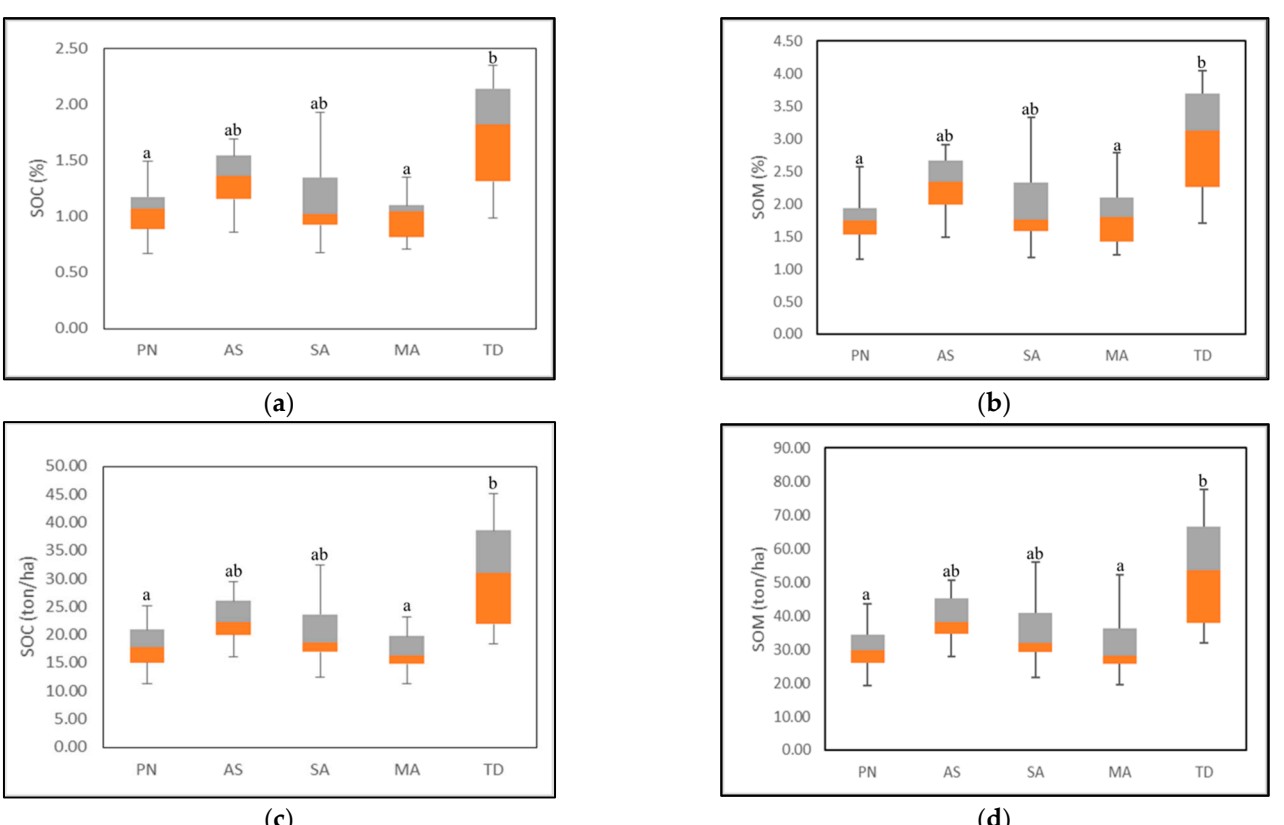

**Figure 2.** Variations in SOC and SOM among stand types; (**a**) SOC concentration, (**b**) SOM concentration, (**c**) SOC stock and (**d**) SOM stock. Yellow and grey areas of the boxes indicate the second and third quartiles, respectively. Whiskers imply the upper and the lower quartile. Different letters (a,b,c) demonstrate significant differences between plot groups (Annova, *p* < 0.05).

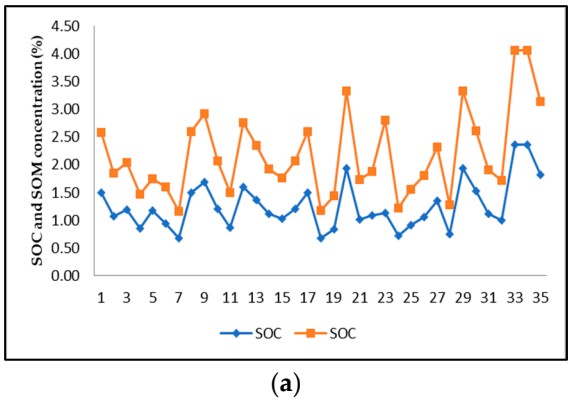

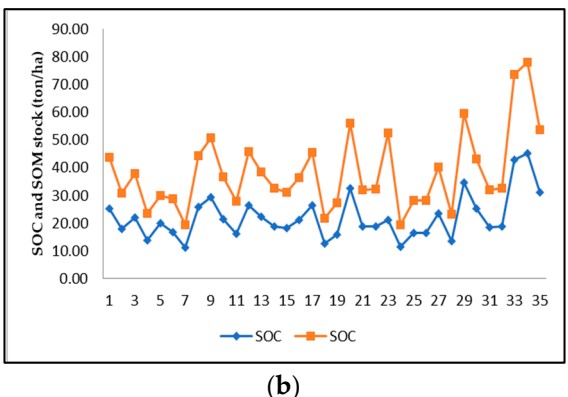

| (**a**) | (**b**) |

**Figure 3.** Between plot variations in SOC and SOM; (**a**) concentration (%), (**b**) stock (ton/ha). Blue lines indicate SOC, and yellow lines indicate SOM.

### 3.2.3. Influencing Factors of SOC

The highlighted factors (five tree vegetation factors and 12 others) varied among plots with medium coefficient variation, except soil pH and BD, which were weak variations (Appendix A Figure A1). However, as the common nature of ecological data, there was inter-correlation (multicollinearity) among several predictor variables. Based on the PCA, the highlighted variables were grouped into six factors/components (Table 4).

**Table 4.** Summary statistic of Principal component analysis (PCA).

| NO | Variables | PCA Component | | | | | |
|----|-----------|------|------|------|------|------|------|
| | | 1 | 2 | 3 | 4 | 5 | 6 |
| 1 | Density | −0.130 | **0.878** | 0.161 | 0.075 | −0.112 | −0.199 |
| 2 | Basal Area | 0.106 | **0.785** | −0.181 | 0.224 | 0.076 | 0.253 |
| 3 | Canopy cover (%) | 0.319 | 0.468 | 0.212 | **0.490** | −0.105 | 0.157 |
| 4 | Canopy height (m) | −0.489 | 0.070 | −0.104 | **0.694** | 0.016 | 0.053 |
| 5 | Diversity index | **0.531** | −0.292 | 0.272 | 0.141 | 0.372 | 0.364 |
| 6 | Altitude (masl) | **0.829** | 0.117 | −0.077 | −0.094 | −0.110 | −0.066 |
| 7 | Slope (%) | 0.179 | 0.044 | −0.195 | −0.141 | **0.734** | −0.051 |
| 8 | Slope position (Topography) | **0.681** | −0.356 | −0.196 | −0.137 | 0.215 | −0.191 |
| 9 | Bare Rock (%) | −0.342 | 0.372 | 0.085 | −0.430 | 0.151 | **0.477** |
| 10 | Distance from road (m) | −0.393 | −0.142 | 0.190 | 0.213 | **0.602** | 0.298 |
| 11 | Distance from river (m) | **0.767** | 0.107 | −0.279 | 0.045 | 0.310 | 0.167 |
| 12 | Below-stand utilization | 0.037 | −0.210 | −0.039 | **−0.827** | −0.173 | 0.029 |
| 13 | Soil Texture (Silt) | −0.163 | 0.412 | **0.710** | 0.342 | −0.012 | −0.202 |
| 14 | Soil Texture (Clay) | 0.159 | −0.030 | **−0.963** | 0.030 | −0.057 | −0.005 |
| 15 | Soil Texture (Sand) | −0.054 | −0.399 | **0.628** | −0.412 | 0.095 | 0.225 |
| 16 | pH | 0.015 | 0.003 | −0.043 | 0.019 | 0.001 | **0.906** |
| 17 | BD | 0.107 | −0.039 | 0.171 | 0.141 | **0.711** | 0.006 |
| Eigenvalue | | 3.527 | 2.732 | 2.333 | 1.688 | 1.529 | 1.014 |
| % of Variance explained | | | 20.75 | 16.07 | 13.72 | 9.93 | 8.99 |
| Cumulative % of variance explained | | | 20.75 | 36.82 | 50.54 | 60.47 | 69.46 |

Note: Bold letter indicated members of each component.

Implicit in the PCA results (Table 4), Component 1 consisted of diversity index, elevation and distance from streams where the three factors were positively correlated to each other. Component 2 grouped density and basal area where both are positively related to one another. Furthermore, component 3 was a combination of silt, clay and sand (soil texture), showing that clay is negatively correlated with silt and sand concentration in the composition. Meanwhile, component 4 was a summarized factor of canopy cover, canopy height and below-stand utilization, where canopy cover and canopy height positively related to each other, but both were negatively correlated with below-stand utilization.

Moreover, component 5 grouped slope, distance from road and BD, where slope, distance from road and BD were positively related to each other. Lastly, the sixth component was the mixture factors of bare rock and soil pH, where both are directly proportional.

The RDA exhibited the effect of the six components on SOC and SOM. As shown in Table 5, the RDA showed 0.229 of the eigenvalue in the first axis and showed 28.8% of the total variance explained. Interpreted from the first axis (Table 5; Figure 4), component 4 showed a moderate positive correlation to SOC concentration, SOC stock, SOM concentration and SOM stock. Other components showed weak correlations (components 2, 3 and 6) and no significant correlations (components 1 and 5). Therefore, component 4 provided the most dominant influence on SOC in this study. Based on the variation partitioning, component 4 contributed almost half of the total variance explained as this factor had 13.20% of the variance explained, while the other components hold only 1.6%, 4.4%, 4.7%, 0.5% and 4.4% for components 1, 2, 3, 5 and 6, respectively (Table 5).

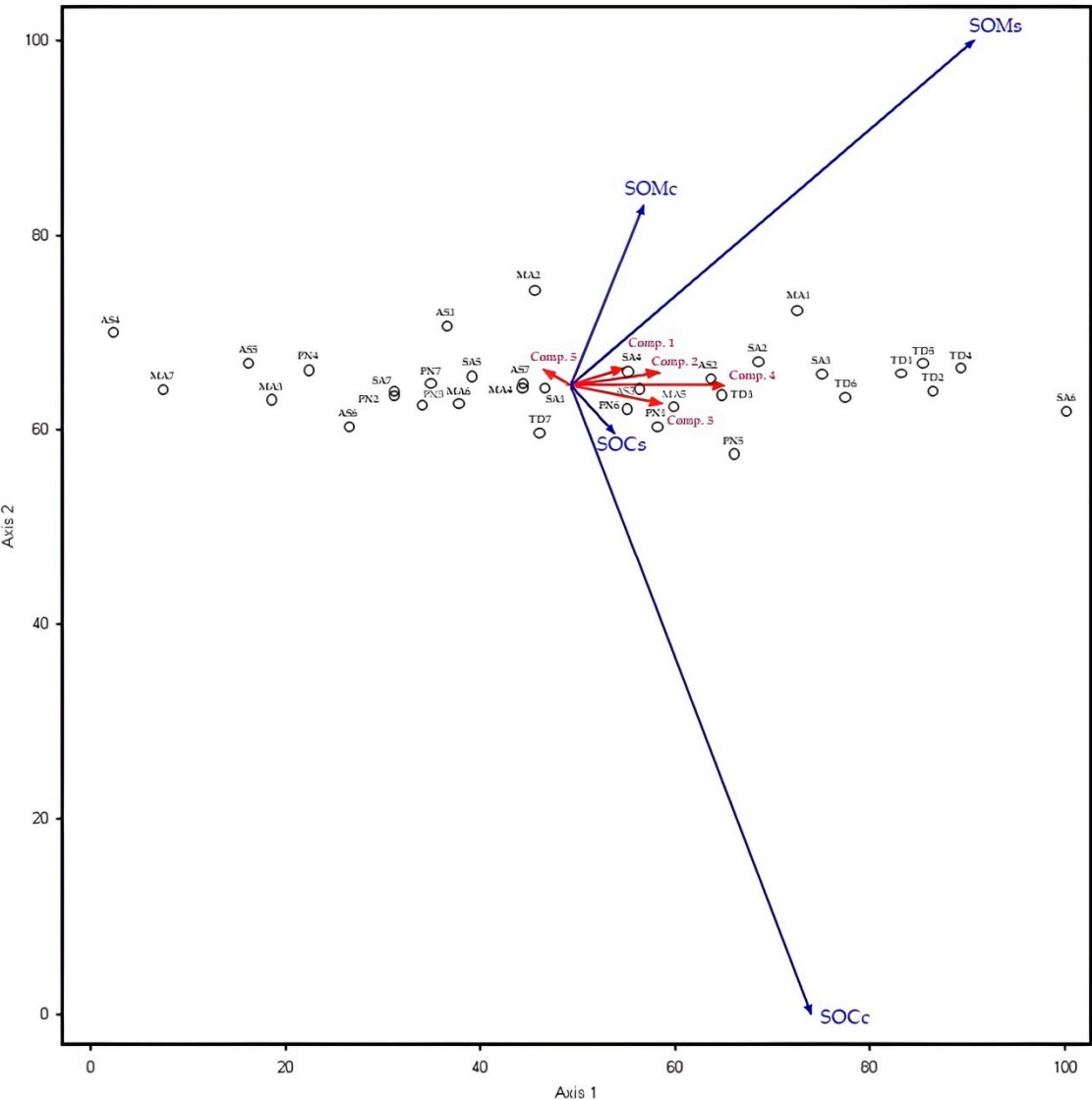

**Figure 4.** Effect of tree vegetation characteristics and environmental factors on the concentration and stock of SOC and SOM in the protected forest of Kulon Progo Community Forestry. SOCc is SOC concentration, SOMc is SOM concentration, SOCs is SOC stock and SOMs is SOM stock.

**Table 5.** Summary Statistics of Redundancy Analysis (RDA) between tree vegetation communities and environmental factors.

|  | Axis1 | Axis2 | Axis3 | Partial Variation (%) |
|---|---|---|---|---|
| Summary statistic: | | | | |
| Eigenvalues | 0.229 | 0.002 | 0.000 | - |
| Variance Explained (%) | 28.500 | 0.200 | 0.000 | - |
| Cumulative explained (%) | 28.500 | 28.700 | 28.800 | - |
| Pearson correlation | 0.519 | 0.517 | 0.652 | - |
| Inter-set correlation: | | | | |
| Component 1 | 0.123 | 0.300 | −0.142 | 1.60 |
| Component 2 | 0.210 | 0.214 | −0.115 | 4.40 |
| Component 3 | 0.215 | −0.294 | 0.288 | 4.70 |
| Component 4 | 0.365 | 0.01 | 0.081 | 13.20 |
| Component 5 | −0.067 | 0.290 | 0.524 | 0.50 |
| Component 6 | 0.211 | 0.002 | −0.069 | 4.40 |

## 4. Discussion

### 4.1. Tree Species Diversity and Structural Characteristics

Differences in tree vegetation composition among all groups (Appendix A Table A1) were not followed by the same pattern in terms of species diversity (Table 3). In addition, due to the high dominance of several species, species diversity in the entire stand types was generally low category as H′ values in all stand types was <2.30 [67]. However, at the tree layer, the PN stand showed lower H′ compared to other stand types because the tree density in PN was more concentrated to one species (*P. merkusii*). The density of stands concentrated in certain species contributes to the low value of H′ because the proportion of the number of individuals of each species is very decisive in calculating H′ [59]. Accordingly, plant enrichment at the lower layers (smaller growth level) will lead to a smaller variation of H′ among stand types. Plant enrichment in this area has been gradually applied by planting some tree species producing NTFs since this forest area was designated as a protected forest in 2007 [46]. In a non-production forest, community income from forests depends on non-timber forest products [35,36] and environmental/ecosystem services [32]. Furthermore, we then found no significant difference in H′ among stand types at the sub-tree layer (pole) and shrub layer (sapling), although these values were still in the low category (Table 4).

In contrast to the density and basal area, PN had a comparable average canopy height with SA, MA and TD and even significantly showed higher canopy height than AS. All the existing dominant tree species, including *P. merkusii, A. molucana, S. macrophylla, M. leucadendra, A. mangium, T. grandis* and *D. latifolia,* were tall trees with a max height of more than 20 m [68,69]. Therefore, the lower canopy height in AS might be coincidently related to age and local environmental conditions because the planting of tree species in the study site mostly depended on farmer preference. Age is closely related to tree height, and both are often used as measures of site quality [70]. Moreover, canopy density/canopy cover showed different patterns, where the TD stand had a significantly denser canopy cover than PN and MA. In the TD stand, the denser canopy cover gained from the combination of *T. grandis* and *D. latifolia*. Widely known that *T. grandis* is a broad-leaved tree with a wide canopy [68,69], and *D. latifolia* has a dome-shaped canopy with lush green foliage [69]. Inversely, there were more canopy gaps in PN because *P. merkusii* is a needle-leaved tree, although it has a wide and thick canopy [68]. Meanwhile, the combination of *A. auriculiformis* with a rounded canopy [69] and *M. leucadendra* with small leaves [68] also provides less canopy cover than the TD stand.

### 4.2. Variation in SOC

The protected forest of Kulon Progo Community Forestry had low SOC and SOM because the average SOC concentration in all stand types was only between 1 and 2% (Figure 2). At the plot level, most of the sample plots also showed less than 2% of SOC concentration in the soil samples (Figure 3). Only a few plots located in the TD stand showed moderate values. According to PPT [58], SOC values followed these levels: SOC concentration < 1% = very low, 1%–2% was low, 2%–3% = moderate, 3%–5% = high and >5% = very high.

The low values of SOC concentration might be related to site characteristics dominated by latosol soil with steep slopes and hilly topography [39]. Some previous studies suggested that latosol soil usually contained low SOC. An experiment by Yakti et al. [71] showed that the SOC concentration in latosol soil was only 0.84%, included in SOM content of 1.452%. A similar finding was revealed by Pinheiro et al. [72] that SOC at 5–10 cm soil in red latosol soil in Brazil was only 1.1%. In addition, other soil types found in several spots of the study site were inceptisol and entisol soils which also have generally low SOC contents, especially in dry land such as our study site. This was in-line with a finding of Fazrina et al. [73], revealing that inceptisol and entisol in dry land in the Aceh-Besar district showed only 0.4%–0.72% and 0.15%–1.25% SOC concentrations, respectively. Furthermore, López-Hernández et al. [74] found that entisol soil had the lowest SOC concentration in an open area (savanna) in South Africa, with only 0.54%. Therefore, considering the site characteristic, the average SOC concentration between 1 and 2% in the protected forest of Kulon Progo Community Forestry was acceptable and did not reflect the absence of the influence of vegetation and environmental factors on SOC.

#### 4.2.1. Effect of Different Vegetation Types on SOC

In general, our finding was consistent with many previous studies that vegetation increases SOC [15,16], although the influences maybe related to other environmental factors [11,17,18]. TD stands hold the highest SOC in both concentration and stock (Figure 2), which was consistent with the better condition in tree structural characteristics (Table 3). Although TD was actually having comparable basal area and density to other stand types, especially in the tree layer, the character of *D. latifolia* with a lush and dense crown [68,69] and *T. grandis* with a wide crown and broad leaves [69] resulted in wider/denser canopy cover (Table 4). According to You et al. [22], vegetation community structure may affect the SOC pool by changing the microenvironment and soil characteristics. Therefore, it was reasonable that AS and SA stands showed equal values SOC to TD following their similarity in structural characteristics, including diversity index, basal area, density and canopy cover (Table 3). Like TD stand, AS was also characterized by a wide canopy from *A. molucana* [75] and *Swietenia macrophilla* [76]. In addition, these two species had strong rooting systems [75,76]. Therefore, the combination of these species in AS might contribute to the equal value of SOC between AS and TD. Moreover, the combination of the most dominant species in SA, including *Acacia auriculiformis* with rounded canopies [68] and *Swietenia macrophilla* with a wide canopy [76], likely provided a good environment as in TD and AS.

In similar environmental factors such as climate, topography and soil type of the entire study site, *D. latifolia*, *A. molucana* and *A. auriculiformis* indicated great potential in SOC sequestration and seemed to be determinant factors of the higher SOC in TD, AS and SA stands, respectively. The average SOC in TD (SOC concentration 1.72%, SOC stock 30.82 ton/ha) was in line with some previous studies and even indicated a better result, especially when considering the latosol soil in the study site. Previous studies suggested that monoculture *T. grandis* stand had lower SOC concentration than other stands in similar environmental factors in Central Java, especially compared to mixed stands [77]. Research conducted by Riniarti and Setiawaan [78] revealed that soil under tree stands dominated by *D. latifolia* contains higher SOC than other stand types. Meanwhile, monoculture stands of *T. grandis* in several community forests in Central Java were reported to contain less SOC than

mixed stands [77]. Therefore, the mixing of *D. latifolia* and *T. grandis* in TD stand seemed to have a useful effect on SOC in this area. According to Devi [11], mixing tree species provide a significant effect on SOM and SOC distribution. Moreover, the average SOC concentration of 1.33% in AS and 1.17% in SA were equal to the SOC concentration explored in some private community forests, which are extensively cultivated. Suyana et al. [77] reported SOC concentration of mixed private community forest in a hilly region of Central Java was 1.12%. Another finding was reported by Pham et al. [79] that *A. auriformis* and *A. mangium* forest plantations in mountainous regions of Central Vietnam exhibited ranges of SOC concentration from 0.45% to 1.85%.

Contrasting conditions appeared in PN and MA, which significantly showed lower SOC compared to TD. In PN, it was strikingly different, with some previous papers revealing that coniferous forests usually accumulate greater SOC than broad-leaves forests in temperate regions [11,80]. Coniferous forests were usually grown or planted in areas with lower temperatures associated with elevation [11], thus supporting the delay of litter decomposition, allowing SOC to accumulate [81]. In temperate forests, coniferous forests are also considered as having a dense and low/short canopy, which inhibits raindrops and reduces the rate of erosion [13,24]. However, in tropic regions, especially in Indonesia, coniferous forest (*P. merkusii*) is a tall tree and commonly a monoculture plantation which is also considered low SOC accumulation [82]. Moreover, *P. merkusii* usually requires a higher elevation and annual rainfall compared to species such as *D. latifolia* and *T. grandis* in TD. Normally, *P. merkusii* grows in areas of high rainfall above 2000 mm/year in elevations of 400–2000 masl [83]. In fact, PN and other stand types in our study were located in the same cluster of hills with similar temperatures and annual rainfall [39]. In addition, pine species with a sparse canopy cover of coniferous leaves and high water consumption lead to high evapotranspiration [84,85], soil drying and high soil temperature [83]. The higher temperature has been reported as negatively impacting SOC accumulation and sequestration [13,81]. Another study showed that *Pinus merkusii* stands in Merbabu mount contained lower SOC levels compared to other stands [29].

In the MA stand, a similar mechanism to PN was possible to occur as the most dominant tree species (*M. leucadendra*) in MA showed less density compared to PN (Table 3) and was categorized as a small canopy [86]. The lower density of *M. leucadendra* was the common feature of the *M. leucadendra* plantation in all community forestry in Kulon Progo, which is categorized as low [39]. In terms of canopy cover, *M. leucadendra* is characterized by a straight stem, little and irregular branching, small leaves and a narrow crown [69,86].

4.2.2. Main Influencing Factors

Analysis of the effect of tree vegetation characteristics and some associated factors confirmed the variation of SOC among stand types. RDA showed that the six grouped factors (components) influenced SOC, with 28.8% of the total variance explained (Table 5). Component 4 (canopy height, canopy cover and below-stand utilization) was the most influential factor on both SOC and SOM (Table 5; Figure 4). This component accounted for nearly half of the total variance explained (Table 5). Our finding was in-line with some previous studies in Indonesia. Muardimansyah et al. [87] reported that tree stands dominated by denser canopies contained higher SOC in a protected forest in Donggala, Central Sulawesi. Similar results were also implied in a study conducted by Suyana et al. [29], revealing that SOC in Merbabu Mt. National Park also was more stored under the denser tree stands. A study in China also suggested that vegetation with denser canopies prevents nutrient loss, including SOC [88]. According to Wenjie et al. [84] and Fan et al. [85], the high and dense canopy cover reduces evaporation from the land or forest floor. Trees with wide and dense canopies also generally have large and deep roots resulting in high soil moisture obtained from the "hydraulic lift" process [89]. Furthermore, the cold temperature under the dense canopy cover is able to maintain and increase SOC sequestration [11,81]. In addition, a high and dense canopy also reduces the direct impact of raindrops on the ground reducing the rate of erosion [13,24]. According to Rosose et al. [90] and Wang et al. [3], the rate of erosion

influenced the loss of SOC and soil fertility. Moreover, a decrease in below-stand utilization in a high and dense canopy will further reduce the risk of erosion [11] and provide a benefit for the accumulation of litter and understorey cover, increasing the supply of SOM and SOC [23,81].

Other grouped factors (components) showed smaller contributions in affecting SOC, although particular factors are sometimes given higher influence in certain areas. For instance, Suyana et al. [76] found that density indicated a positive effect on SOC. The small effect of density and basal area (component 2) in this study was likely caused by differences in vegetation characteristics among plots or by other stronger factors. For instance, the greater density and basal area in PN were not followed by greater SOC and SOM due to the character of P merkusii, which has a sparse canopy. In addition, *P. merkusii* is also known as a high water consumer, resulting in high evaporation and evapotranspiration [84,85]. Meanwhile, in component 3, soil texture in the whole area of our study site was relatively similar to the balanced composition of sand, silt and clay (loam texture). Only a few plots showed different soil textures, namely clay-loam textures. However, our findings were quite enough to confirm the results of several studies that soil texture has an effect on SOC. We found that some plots with higher clay content had lower SOC. This is in accordance with articles published by Devi [11] and Suyana et al. [29], reporting that clay concentration negatively affects SOC. Furthermore, soil pH in component 6 was also reported to have a low correlation with SOC [13].

The diversity index in component 1 was categorized as a low diversity index for all plots and all stand types, so it only provided a small contribution. In addition, the elevation and the distance from the river in this study had only a small range resulting in a smaller contribution. In various stands with high ranges of elevation, SOC is possibly more influenced by elevation [12]. Moreover, slope, distance from the road and BD grouped in component 5 also showed only a small range, so they provided the smallest contributions to the total variance.

## 5. Conclusions

Tree vegetation affects SOC, although the influences are often related to other environmental factors. Our result found significant differences in tree vegetation composition and characteristics, especially in the tree layer. As we hypothesized, the difference pattern seems to be directly proportional to the variation of SOC and SOM values. RDA confirmed that tree vegetation held an important role in SOC and SOM storage in the protected forest of Kulon Progo community forestry, although it was also related to other associated factors. Lower SOC and SOM were explored in Pinus and Melaleuca-Acacia stands, while the greater ones were found in Tectona-Dalbergia, Aleurites-Swietenia and Swiietenia-Acacia stand types. The most influential factor for SOC and SOM storage and accumulation was a combination factor of canopy cover, canopy height and below-stand utilization, where canopy cover was directly proportional to canopy height and increased with decreasing below-stand utilization. In other words, the higher the canopy cover and canopy height, the lower the below-stand utilization and ultimately increase the SOC storage. Accordingly, dense-canopy trees are highly recommended for further forest management. The SOC storage and accumulation will be greatly conditioned through the development of such species, followed by adaptive management of below-stand utilization.

Our findings provide fundamental information for maximizing the potential of forest carbon to meet the global payments for ecosystem services and to increase the forest contribution to low carbon development strategies and emission reductions from forests and other land use sectors (FOLU) as targeted in the Nationally Determined Contributions (NDC) of Indonesia. Repetition of this study in other protected forest areas with various environmental conditions related to the role of tree vegetation was suggested to enrich this finding.

**Author Contributions:** Conceptualization, S.; methodology, S. and C.-W.Y.; software, S., H.K. and J.L.; validation, S. and C.-W.Y.; formal analysis, S.; investigation, S., H.K. and J.L.; resources, C.-W.Y.; data curation, S.; writing—original draft preparation, S.; writing—review and editing, S.; visualization, S., H.K. and J.L.; supervision, C.-W.Y.; project administration, J.L.; funding acquisition, C.-W.Y. All authors have read and agreed to the published version of the manuscript.

**Funding:** This work was supported by the research grant of Kongju National University in 2022.

**Data Availability Statement:** Data are contained within the manuscript.

**Acknowledgments:** The authors would like to thank Bambang Dwi Atmoko, Lathif Brahmantya, Eko Priyanto and Uchu Waluya Heri Pahlana for their valuable support in data collection. The authors also thank Susi Abdiyani for providing a careful proofread of the manuscript, especially for English style, spelling and grammar.

**Conflicts of Interest:** The authors declare no conflict of interest.

## Appendix A

**Table A1.** Species Composition.

| Tree Layer | | Sub-Tree Layer | | Shrub Layer | |
|---|---|---|---|---|---|
| **Species** | **IV (%)** | **Species** | **IV (%)** | **Species** | **IV (%)** |
| PN | | | | | |
| *P. merkusii* | 70.14 | *P. merkusii* | 22.16992 | *D. latifolia* | 16.97593 |
| *A. camansi* | 6.17 | *E. cyclocarpum* | 20.63984 | *A. heterophyllus* | 15.3369 |
| *M. leucadendra* | 6.11 | *A. camansi* | 13.78127 | *S. macrophylla* | 14.39476 |
| *T. grandis* | 2.80 | *G. genemon* | 12.60789 | *D. zibetinus* | 10.27393 |
| *Swietenia* sp | 2.34 | *A. Altilis* | 8.547447 | *P. speciosa* | 6.898614 |
| *E. cyclocarpum* | 5.26 | *M. Leucadendra* | 7.871581 | *A. Altilis* | 6.42607 |
| *A. altilis* | 2.50 | *P. speciosa* | 7.871581 | *L. leucocephala* | 6.318482 |
| *P. speciosa* | 4.70 | *A. heterophyllus* | 6.51 | *S. aromaticum* | 5.843828 |
| - | - | - | - | *S. densiflora* | 5.843828 |
| - | - | - | - | *H. brasiliensis* | 5.84 |
| - | - | - | - | *E. cyclocarpum* | 5.843828 |
| Total | 100.00 | | 100.00 | | 100.00 |
| AS | | | | | |
| *A. molucana* | 52.39 | *S. Macrophylla* | 34.40032 | *Swietenia* sp. | 30.63431 |
| *Swietenia* sp. | 21.49 | *M. Leucadendra* | 16.24192 | *A. mangium* | 11.62835 |
| *A. mangium* | 15.05 | *T. grandis* | 13.48878 | *A. heterophyllus* | 11.03645 |
| *Eucalyptus* sp. | 5.90 | *P. speciosa* | 11.92149 | *T. cacao* | 9.812265 |
| *P. speciosa* | 2.66 | *A. heterophyllus* | 11.64972 | *T. grandis* | 6.419125 |
| *M. Leucadendra* | 2.51 | *A. camansi* | 6.21713 | *P. speciosa* | 6.419125 |
| - | - | *A. Auricuiformis* | 6.080649 | *M. Leucadendra* | 5.79059 |
| - | - | - | - | *J. curcas* | 5.245859 |
| - | - | - | - | *D. zibetinus* | 4.784933 |
| - | - | - | - | *G. genemon* | 4.114495 |
| - | - | - | - | *G.eliptica* | 4.114495 |
| Total | 100.00 | | 100.00 | | 100.00 |

**Table A1.** *Cont.*

| Tree Layer | | Sub-Tree Layer | | Shrub Layer | |
|---|---|---|---|---|---|
| **Species** | **IV (%)** | **Species** | **IV (%)** | **Species** | **IV (%)** |
| SA | | | | | |
| *A. mangium* | 30.78 | *S. macrophylla* | 26.15 | *Swietenia* sp. | 28.13 |
| *S. Macrophylla* | 22.86 | *A. mangium* | 15.29 | *D. latifolia* | 21.89 |
| *D. latifolia* | 11.52 | *D. latifolia* | 14.43 | *A. mangium* | 10.84 |
| *T. grandis* | 10.89 | *M. leucadendra* | 13.47 | *C. calothyrsus* | 7.66 |
| *M. Leucadendra* | 6.45 | *T. grandis* | 11.83 | *M. Leucadendra* | 6.21 |
| *E. cyclocarpum* | 3.87 | *P. speciosa* | 4.93 | *P. speciosa* | 6.16 |
| *P. merkusii* | 2.80 | *A. molucana* | 3.14 | *A. heterophyllus* | 6.13 |
| *Eucalyptus* sp. | 2.54 | *P. falcataria* | 3.09 | *T. grandis* | 4.82 |
| *P. speciosa* | 1.91 | *A. heterophyllus* | 2.72 | *Eucalyptus* sp. | 1.55 |
| *C. nucifera* | 1.48 | *A. Altilis* | 1.36 | *M. indica* | 1.55 |
| *S. cumini* | 1.46 | *Eucalyptus* sp. | 1.26 | *G. genemon* | 1.55 |
| *A. molucana* | 1.05 | *A. camansi* | 1.26 | *A. elliptica* | 1.30 |
| *A. camansi* | 0.99 | *G. genemon* | 1.06 | *D. zibetinus* | 1.20 |
| *L. Leucocephala* | 0.70 | - | - | *J. curcas* | 1.01 |
| *A. heterophyllus* | 0.70 | - | - | - | - |
| Total | 100.00 | | 100.00 | | 100.00 |
| MA | | | | | |
| *M. Leucadendra* | 39.80 | *M. Leucadendra* | 27.22 | *A. mangium* | 16.79 |
| *A. mangium* | 24.07 | *P. speciosa* | 24.91 | *P. speciosa* | 13.78 |
| *Eucalyptus* sp. | 8.63 | *A. mangium* | 18.02 | *M. indica* | 11.19 |
| *T. grandis* | 8.46 | *S. Macrophylla* | 9.51 | *Swietenia* sp. | 10.63 |
| *A. molucana* | 6.31 | *P. canescens* | 6.06 | *A. heterophyllus* | 10.14 |
| *D. latifolia* | 3.73 | *P. falcataria* | 2.95 | *D. latifolia* | 7.64 |
| *S. Macrophylla* | 3.70 | *D. latifolia* | 2.95 | *D. zibetinus* | 6.95 |
| *P. speciosa* | 1.77 | *Eucalyptus* sp. | 2.83 | *M. Leucadendra* | 6.26 |
| *P. merkusii* | 1.77 | *A. molucana* | 2.83 | *G. genemon* | 6.26 |
| *P. canescen* | 1.77 | *A. pauciflorum* | 2.72 | *S. aromaticum* | 5.82 |
| | - | - | - | *G. sepium* | 4.54 |
| Total | 100 | | 100 | | 100 |
| TD | | | | | |
| *D. latifolia* | 36.89 | *D. latifolia* | 30.95 | *D. latifolia* | 50.03 |
| *T. grandis* | 34.95 | *S. Macrophylla* | 26.98 | *Swietenia* sp. | 17.14 |
| *S. Macrophylla* | 9.56 | *M. Leucadendra* | 19.17 | *T. grandis* | 10.33 |
| *M. Leucadendra* | 6.04 | *T. grandis* | 15.95 | *L. leucocephala* | 7.31 |
| *A. mangium* | 3.52 | *G. genemon* | 1.94 | *A. heterophyllus* | 4.79 |
| *E. cyclocarpum* | 3.10 | *Eucalyptus* sp. | 1.87 | *G. genemon* | 4.46 |
| *Eucalyptus* sp. | 2.83 | *A. mangium* | 1.57 | *G. sepium* | 2.57 |
| *A. molucana* | 1.50 | *A. molucana* | 1.57 | *M. Leucadendra* | 2.01 |
| *L. Leucocephala* | 0.83 | - | - | *P. speciosa* | 1.37 |
| *P. speciosa* | 0.78 | - | - | - | - |
| Total | 100 | | 100 | | 100 |

Note: PN = *Pinus* stand, AS = *Aleurites-Swietenia* stand, SA = *Swietenia-Acacia* stand, MA = *Meulaleuca-Acacia* stand, TD = *Tectona-Dalbergia* stand. IV = important value.

**Table A2.** Summary statistics of multi-response permutation procedure (MRPP) analysis for tree vegetation communities.

| No | Comparison of Sorensen Distance | T | A | *p*-Value |
|---|---|---|---|---|
| 1 | General Comparison | −28.27 | 0.32 | 0.000 |
| 2 | Pairwise Comparison: | | | |
| | *1* vs. *2* | −15.47 | 0.32 | 0.000 |
| | *1* vs. *3* | −12.74 | 0.22 | 0.000 |
| | *1* vs. *4* | −11.48 | 0.20 | 0.000 |

**Table A2.** *Cont.*

| No | Comparison of Sorensen Distance | T | A | *p*-Value |
|---|---|---|---|---|
| *1* vs. *5* | | −13.50 | 0.13 | 0.000 |
| *2* vs. *3* | | −11.07 | 0.49 | 0.000 |
| *2* vs. *4* | | −8.85 | 0.46 | 0.000 |
| *2* vs. *5* | | −16.35 | 0.28 | 0.000 |
| *3* vs. *4* | | −9.36 | 0.26 | 0.000 |
| *3* vs. *5* | | −12.54 | 0.15 | 0.000 |
| *4* vs. *5* | | −9.84 | 0.13 | 0.000 |

Note: T = separation between groups, A = within-group homogeneity, p = significance level at alpha 0.05, UF = undisturbed post-logged forest, JF = jungle rubber forest, MF = mixed regrowth forest, NF = newly regrowth forest (open area).

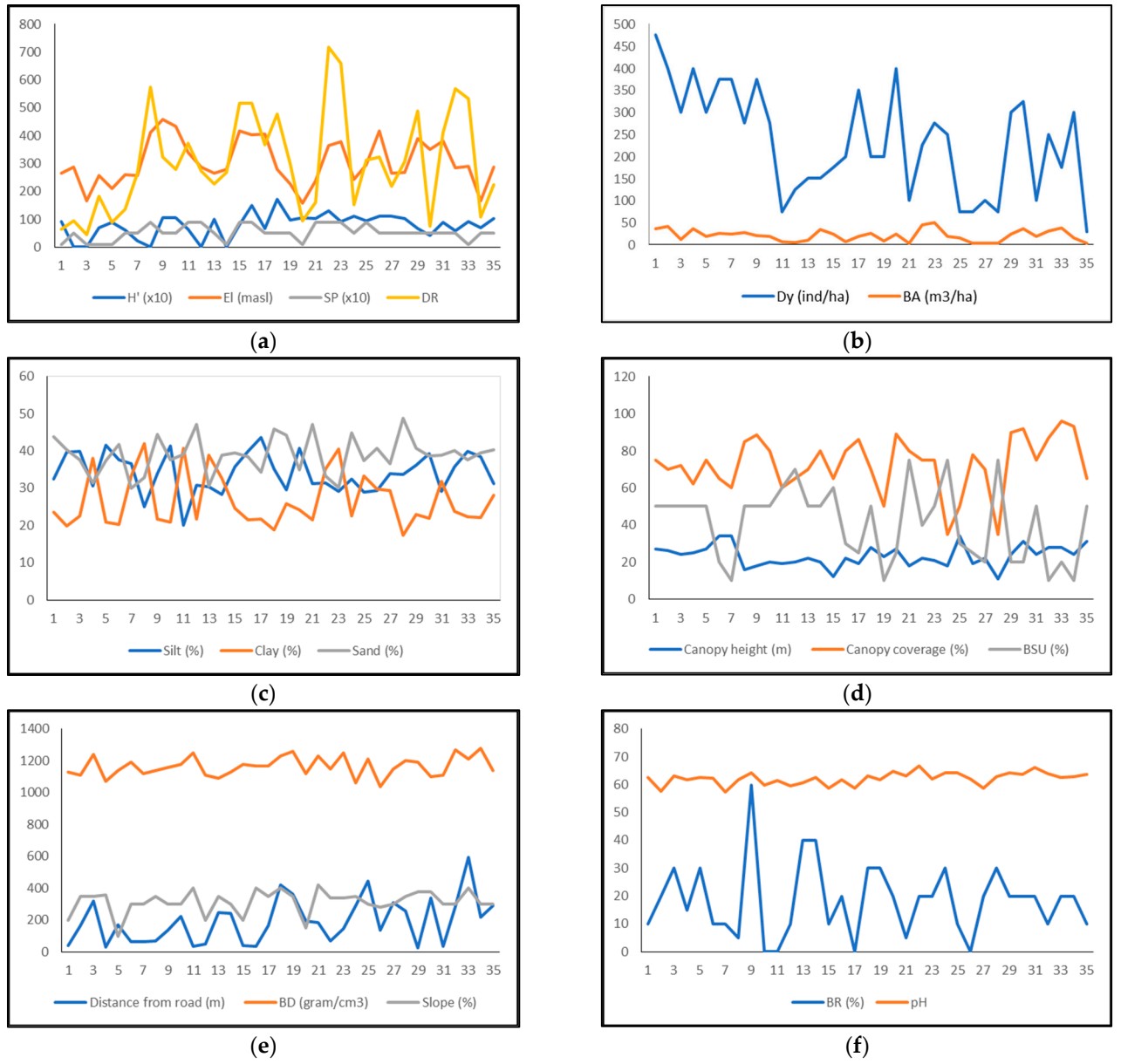

**Figure A1.** Relationship graphs of vegetation characteristics and associated factors within PCA components (**a**) = component 1, (**b**) = component 2, (**c**) = component 3, (**d**) = component 4, (**e**) = component 5 and (**f**) = component 6.

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
