# Peer review of "Influence of Tree Vegetation and The Associated Environmental Factors on Soil Organic Carbon; Evidence from “Kulon Progo Community Forestry,” Yogyakarta, Indonesia"

_forests, doi:10.3390/f14020365_

Round 1
Reviewer 1 Report
The topic presented by the authors is very interesting, relevant to efforts to deal with climate change mitigation, and following the FORESTS scope. Manuscripts have been arranged systematically. A few things still need to be improved:
L 93: Missing title of section 2.
L 142: There were 72 plots for five stand types. Please describe the number of plots for each stand type.
L180-182: Please add citations/references.
L 238: Add notes/remarks for all abbreviations in column “Tree Characteristics”.
L 426: Based on the eight main species listed in Table 1, authors should add a review in this section about single species with the greatest potential to contribute SOC under the same environmental factors (climatic, topographic, and edaphic factors).
L 678: In the appendix (A1), the authors should list all the tree species names found on each stand type.
L 689: literature number [40] has been cited several times in this manuscript, but it is unclear whether this reference is a book, journal article, report, thesis, ….. etc.
Author Response
Dear Reviewer
Thank you very much for your review. We have revised our manuscript according to your comments. Please see the attachment.

Reviewer 2 Report
The manuscript is well-prepared, the experimental design is acceptable, the result analysis and discussion are reasonable. However, some minor issues underlined below need to be taken under consideration before publication.
For the introduction as well as rest of the manuscript, English language and grammar should be cheked and proofread. For example:
Line 31 – should be soil organic matter
Lines 39-41- doesn’t make sense
Line 52- “is” should be deleted, etc.
Author Response
Dear Reviewer
Thank you very much for your review
We have revised our manuscript according to your comments and suggestion.
Please see the attachment.

Reviewer 3 Report
This study was designed to evaluate the Influence of tree vegetation and the associated environmental factors on soil organic carbon; evidence from “Kulon Progo Community Forestry”, Yogyakarta, Indonesia. The study was well designed and carefully conducted, and the results are addressing important questions about carbon development strategies and emission reduction.
I have same major concerns regarding the suitability of publication for this manuscript.
My first major concern is that the authors do not give any hypothesis. Based on the summary of previous studies, authors should clearly present the aims and hypothesis of this study and discuss hypothesis in the “Discussion”.
My second major concern is that the “Conclusions” section of the current manuscript is not well written and needs to improve. the "Conclusions" is too long now. Authors could setup 2-3 subtitles in the “Conclusions” section to summarize the new findings of this manuscript.
Author Response
Dear Reviewer
Thank you for your review
We have revised our manuscript according to your comments
Please see the attachment.

Reviewer 4 Report
Manuscript “Influence of tree vegetation and the associated environmental factors on soil organic carbon; evidence from “Kulon Progo Community Forestry”, Yogyakarta, Indonesia” tried to assess the influence of tree vegetation and some key environmental factors on soil organic carbon (SOC) in 5 stand types in the protected forest of Kulon Progo Community Forestry, which has an important meanings for vegetation recovery in the protected areas. There are some suggestions for this manuscript to improve.
(1) The Introduction part needs to provide more sufficient background and describe more importance of this study.
(2) “Vegetation Sampling and Survey” part should add a table to state the experimental design and the added more details compared to the research of Siswo ‘s.
(3) In “Statistical Analysis” part, I suggest the relevant model formula and detailed factors should be showed in the manuscript more clearly.
So, I suggest this manuscript should be moderate revison for publish.
Author Response
Dear Reviewer
Thank you for your review
We have revised our manuscript according to your comments and suggestion
Please see the attachment.

Round 2
Reviewer 3 Report
No suggestion
Author Response
Once again, thank you very much for reviewing our manuscript and accepting our revision with no more suggestion.